# Protocol: Waiting time and ways of accessing specialized health services in public hospitals in Ecuador

**Marcelo Armijos Briones**●*, **Sammy Figueroa Intriago, Antonio Lanata-Flores**●, **Pablo Benitez Sellán, Oscar Marcillo Toala**●, **Patricia Estefanía Ayala Aguirre**

School of Dentistry, Universidad de Especialidades Espíritu Santo, Samborondón, Ecuador

\* fernandoarmijos@uees.edu.ec

## Abstract

This study aims to determine the waiting time and the forms of access to specialized health services in public hospitals in Ecuador. A representative sample of 32 hospitals under the Ministry of Public Health was considered, with 26 selected by accessibility convenience. Data will be collected using a structured questionnaire. Patients will be asked about the number of days they waited for their medical appointments and the method used to schedule their appointments. The study distinguishes between standardized access, based on Ecuador's formal referral and counter-referral system, and non-standardized access methods, such as personal connections or hospital staff involvement. The data of this protocol are registered and publicly accessible at: https://dx.doi.org/10.17504/protocols.io.261ge5z7wg47/v1. We expect to identify a correlation between waiting times and the type of access to specialized medical services, with non-standardized access potentially leading to shorter waiting times. This research may highlight disparities in the system and suggest areas for improvement in equity and efficiency within the healthcare referral process. To do so, a structured survey will be used. Since a construct is not needed to determine the waiting time or the forms of access, it was not necessary to validate the instrument. However, we did validate the understanding of the questions and the response options in several places in the country. According to the results of the validation of the instrument, pollsters will be instructed to inform users about the meaning of the question on ethnic identification, which was difficult to understand in the country's coastal areas.

## Introduction

The Ecuadorian health system can be considered as segmented or mixed. This system brings together 5 health subsystems: The public subsystem that serves the population without any type of health insurance is free of charge and administered by the Ministry of Public Health. Then, there is the public health insurance for workers or people in employment relationships, the Ecuadorian Social Security Institute. The Social Security Institute of the National Police (ISSPOL) and the Social Security Institute of the Armed Forces (ISSFA) are health subsystems

**Data availability statement:** All relevant data are within the manuscript and its Supporting Information files. They are also available at https://dx.doi.org/10.17504/protocols. io.261ge5z7wg47/v1[Protocols.io]

**Funding:** FAB; PEA numbers awarded to each author: 10 University Espíritu Santo https:// uees.edu.ec/ the funders had no role in study design, data collection and analysis, decision to publish, or preparation of the manuscript.

**Competing interests:** The authors have declared that no competing interests exist.

exclusive to people who belong to these public forces. Finally, there is the private health sub-system. All of these subsystems have a presence at a national level and serve users at all levels of complexity. These subsystems are linked to each other through the Comprehensive Public Network of Health and Complementary Services. This system aims to cover the needs of the subsystems among themselves and pays the costs of care for patients who do not belong to the subsystem that served them, through a single fee schedule for care [1].

Even though the Constitution recognizes health as a human right in the whole Comprehensive Public Health Network, there are challenges that need to be addressed such as: extensive/long working hours (or shifts), availability and quality of medicines and a perceived disconnect between governance and medical service [1,2]. Despite constitutional guarantees for healthcare access, the dominance of the biomedical model and top-down policy enforcement have led to issues in addressing health epidemics in rural areas [3]. Faith-based healthcare providers (FBHPs) have historically contributed in parallel with the system, but their current role in the national health care requires further understanding [4]. In summary, Ecuador's health system is a complex interplay of the public and private sectors, with constitutional commitments to health as a human right and ongoing efforts to improve public health infrastructure. However, the system faces significant challenges, including resource limitations, healthcare provider burnout, and the need for better integration of various health services and innovative systems. Addressing these issues is crucial for the system's reliability and health outcomes in the Ecuadorian population [5]. For example, to improve system reliability, Ecuador created the national referral and counter-referral subsystem, which, among other things, aims to make specialized care more efficient, leaving the first level of care as the gateway to the national health system and regulating access to specialized health services for those who need it.

The referral and counter-referral system in Ecuador, as described in the context provided, is part of a healthcare framework that was reformed following the 2008 Constitution, which guaranteed access to healthcare for all citizens [3]. This system involves the process of referring patients from lower-capacity institutions to higher-complexity institutions for specialized care and diagnoses and then counter-referring them back to the original institution with specific diagnoses and treatment plans [6]. In summary, the referral and counter-referral system in Ecuador is a critical component of the healthcare process designed to facilitate the movement of patients through different levels of care. However, the effectiveness of the system is hindered by the prevailing healthcare model and administrative practices that do not adequately support continuous and community-based care [3,7].

With this protocol, we intend to test the hypothesis that the way of accessing health services intervenes with the waiting time in public hospitals. There is local evidence that this phenomenon is happening [8], but the national dynamics have not been studied systematically. Inquiry into waiting times and forms of access to specialized medical services in public hospitals in Ecuador is a multifaceted issue. This research aims to discern the relationship between the mode of access, whether through formal channels or informal networks and the duration patients wait for specialized consultations. The context provided by Briones et al. (2024) is particularly relevant, as it directly investigates waiting times in Ecuadorian public hospitals and identifies informal access as a potential factor in scheduling delays [8]. Contradictory findings emerge when considering the broader literature on waiting times for health care services. While some studies suggest that socioeconomic status does not significantly affect waiting times for certain services [9], others indicate that lower socioeconomic status is associated with longer waits [10,11]. In summary, the relationship between access forms and waiting times for specialized medical services in Ecuadorian public hospitals is complex and is influenced by various factors, including informal access networks, socioeconomic status,

and external events. Evidence suggests that while formal processes may lead to longer waiting times, informal networks can expedite access, albeit potentially exacerbating inequities [8]. To fully understand the dynamics at play, it is crucial to consider the broader context of healthcare access and waiting times, as illuminated by the referenced international studies.

Understanding the time that users of public hospitals in Ecuador wait for specialized medical care is crucial for improving the country's healthcare system. This information can help policymakers and healthcare providers identify areas that require improvement and allocate resources more effectively. Previous studies have shown that patients in Ecuador often face long waiting times for medical appointments, with some patients waiting months to see a specialist. However, there is limited research on the specific time that users of public hospitals in Ecuador wait for specialized medical care. There is a lack of research on the time that users of public hospitals in Ecuador wait for specialized medical care and how they access these services. This study aims to fill this knowledge gap by providing detailed information on waiting times and referral systems in Ecuadorian public hospitals.

This study is essential for understanding the challenges faced by the Ecuadorian healthcare system and identifying ways to improve the delivery of specialized medical care. This study provides valuable insights for policymakers and healthcare providers by examining waiting times and referral systems in public hospitals.

Therefore, the present study aimed to determine the access methods and its impact on waiting times for specialized medical care in public health care system in Ecuador. It is hypothesized that patients in public hospitals in Ecuador face longer waiting times for specialized medical care, and the referral and counter-referral systems do not efficiently manage patient appointments.

## Materials and methods

A cross-sectional structured survey was conducted to study standardized and non-standardized access methods and its impact on waiting times among patients who have undergone specialized medical consultations in the 26 public hospitals in Ecuador. We will wait to begin conducting surveys outside public hospitals until the article with the protocol is approved. We hope that this will be in april 2025. The approval obtained by the ethics committee lasts one year, until August 2025. If necessary, it is possible to request an extension. Before filling out the surveys, participants must agree to participate after the interviewer reads them the informed consent that is present in digital form in the Google form. Participation by minors or persons who are not in suitable intellectual capacities will not be accepted, due to this, the consent of legal guardians is not necessary.

### Eligibility criteria

Patients who have undergone specialized medical consultations at any of the 26 public hospitals in Ecuador were eligible for inclusion. Exclusion criteria included patients who are unable to provide informed consent, patients with cognitive or linguist limitations that impair their understanding of the questionnaire, and patients who have undergone specialized medical consultations in private hospitals.

### Sample size

Sample size calculation was performed by the Epi Info software version 7 following parameters: confidence level 95%, margin of error 5%, event prevalence 82% because, according to the Briones 2024 where the number of appointments in public hospitals in the province of Manabí in Ecuador are by standardized scheduling. The number of visits to the Outpatient

Clinic areas, where specialized medical care is provided in the country's public hospitals, was taken from the data of the Ecuadorian Ministry of Public Health on its website [12]. The morbidity care data for the month of August 2023 was taken as a reference whenever data from that year existed; when not, they were taken from the last year available as shown in Table 1.

## Data collection

For data collection, a semi-structured survey consisting of a total of 30 questions was conducted with an average completion of 8 minutes. The survey was divided into 3 main sections: 1) social, demographic and economic information; 2) ethnic and cultural information and 3) access methods and waiting time for specialized medical care. For the first two sections, questions were selected from the Health and Nutrition survey forms carried out periodically by the Institute of Statistics and Census of Ecuador (INEC) [13], as shown in Table 2.

For the third section, specialized procedures such as ultrasounds, x-rays, among others, were also considered for the study. The questions in this section were adapted from the INEC Questionnaire in its national surveys, however, when modified to meet the objective of this research, they underwent a validation process to identify their clarity and consistency. Validation was carried out through a pilot study in three hospitals in the country, one in each geographic region (Coastal, Andean highlands and the Amazon). A total of 322 participants were

**Table 1. Stratification of proportional sampling of users who attend medical consultations of specialties in public hospitals in Ecuador.**

| | Hospital | Number of morbidity care visits in August | Year | Sample |
|---|---|---|---|---|
| 1 | Alfredo Noboa Montenegro General Hospital | 2289 | 2022 | 200 |
| 2 | Ambato General Hospital | 11858 | 2023 | 215 |
| 3 | Dr. Verdi Cevallos Balda General Hospital | 7212 | 2023 | 212 |
| 4 | Enrique Garces General Hospital | 9748 | 2023 | 214 |
| 5 | Esmeraldas South General Hospital – "Delfina Torres de Concha" | 3454 | 2023 | 206 |
| 6 | Francisco de Orellana General Hospital | 3450 | 2022 | 206 |
| 7 | Guasmo South General Hospital | 1908 | 2016 | 196 |
| 8 | Gustavo Dominguez General Hospital | 6423 | 2023 | 212 |
| 9 | Homero Castanier Crespo General Hospital | 1581 | 2016 | 192 |
| 10 | Isidro Ayora General Hospital | 508 | 2021 | 153 |
| 11 | José María Velasco Ibarra General Hospital | 6169 | 2023 | 211 |
| 12 | Julius Doepfner General Hospital | 1772 | 2023 | 195 |
| 13 | Latacunga General Hospital | 10468 | 2023 | 214 |
| 14 | León Becerra General Hospital | 2662 | 2022 | 202 |
| 15 | Liborio Panchana Sotomayor General Hospital | 3527 | 2023 | 206 |
| 16 | Luis Gabriel Davila General Hospital | 3263 | 2023 | 205 |
| 17 | Hospital General Macas General Hospital | 2566 | 2022 | 202 |
| 18 | Marco Vinicio Iza General Hospital | 4969 | 2023 | 210 |
| 19 | General Martin Icaza General Hospital | 3152 | 2022 | 205 |
| 20 | Pablo Arturo Suarez General Hospital | 14388 | 2023 | 216 |
| 21 | Hospital General Puyo General Hospital | 4319 | 2023 | 208 |
| 22 | General Rodriguez Zambrano General Hospital | 1876 | 2023 | 196 |
| 23 | Vicente De Paul General Hospital | 5680 | 2023 | 211 |
| 24 | General Teofilo Davila General Hospital | 5577 | 2023 | 211 |
| 25 | Vicente Corral Moscoso General Hospital | 2187 | 2020 | 199 |
| 26 | General Teaching Provincial Hospital of Riobamba | 6946 | 2023 | 212 |
| Total | | 127952 | | 5309 |

**Table 2. Questions and response options for the sections: social, demographic and economic information and ethnic and cultural information.**

| Questions from the first section in Spanish | Answer options | Questions from the first section in English | Answer options |
|---|---|---|---|
| Edad | Número de años de vida cumplidos hasta la fecha de la encuesta | Age | Number of years of life completed up to the date of the survey |
| Sexo | Hombre o mujer | Sex | Female or male |
| Nivel de instrucción formal alcanzado | Ninguno, primaria, secundaria, tercer nivel o cuarto nivel | Level of formal education achieved | None, primary, secondary, third level or fourth level |
| Cuáles son los ingresos promedios de su hogar (colocar la cifra aproximada de los ingresos de todas las personas que viven en su casa, no es necesario un valor exacto) | Cantidad de dinero en dólares que percibe la familia al mes | What is the average income of your household (insert the approximate figure of the income of all the people who live in your house, an exact value is not necessary) | Amount of money in dollars that the family receives per month |
| Cuantas personas viven en su casa (coloque el número de personas que viven en la vivienda del encuestado) | Cantidad de personas viven de forma permanente en su hogar | How many people live in your house (insert the number of people who live in the respondent's home) | Number of people living permanently in their home |
| Área de residencia | Urbano o rural | Area of residence | Urban or rural |
| Vive en la misma provincia en la que está el hospital | Si o no | Do you live in the same province as the hospital? | Yes or no |
| De ser no¿Por qué no se atendió en su provincia? | No hay el especialista que necesito; este hospital queda más cerca de mi residencia; un médico le agendó en este hospital;otra razón | If not, why were you not treated in your province? | The specialist I need is not available; this hospital is closer to my residence; a doctor made an appointment for me at this hospital; other reason |
| Provincia de residencia | Alguna de las 23 provincias continentales del Ecuador | Province of residence | Any of the 23 continental provinces of Ecuador |
| Desde su residencia hasta el hospital¿Cuánto tiempo le tomó llegar al hospital? (colocar el dato en minutos) | El tiempo en minutos que le tomó llegar hasta el hospital | From your residence to the hospital, how long did it take you to get to the hospital? (insert the data in minutes) | The time in minutes it took to get to the hospital |
| En total¿Cuánto gastó en transporte para llegar al hospital? (colocar el valor en dólares del gasto por persona de ida y vuelta) | El valor en dólares que gastó para llegar al hospital | In total, how much did you spend on transportation to get to the hospital? (insert the dollar value of the round-trip expense per person) | The amount in dollars spent to get to the hospital |
| Section on ethnic and cultural information | | | |
| Autoidentificación étnica (preguntarle al encuestado cuál es su etnia) | Mestizo, blanco, indigena, negro, afroecuatoriano, mulato, montuvio, otro | Ethnic self-identification (ask the respondent what their ethnicity is) | Mestizo, white, indigenous, black, Afro-Ecuadorian, mulatto, montuvio, other |

*(Continued)*

**Table 2.** (Continued)

| Questions from the first section in Spanish | Answer options | Questions from the first section in English | Answer options |
|---|---|---|---|
| Cuál es su lengua materna (explique que se trata de su idioma original, el que usa como primera lengua para comunicarse en casa) | Kichwua; Waorani; Shuar; Achuar; Cha´pala; Awapit; Español; otra | What is your native language (explain that this is your original language, the one you use as your first language to communicate at home) | Kichwua; Waorani; Shuar; Achuar; Cha´pala; Awapit; Spanish; other |
| Escriba la otra lengua | Se escribe el nombre del otro idioma mencionado por el paciente en forma textual | Write the other language | The name of the other language mentioned by the patient is written verbatim |
| ¿Alguien en el hospital con quien haya hablado, conocía su idioma? (se refiere a alguna persona que le informó algo respecto a su cita médica) | Si o no | Did anyone at the hospital you spoke to know your language? (refers to someone who told you something about your medical appointment) | Yes or no |
| ¿Cree usted que su etnia es un obstáculo al momento de agendar una cita? (se busca comprender si durante el agendamiento tuvieron dificultades en el proceso por su lengua o etnia) | Si; no; tal vez | Do you think your ethnicity is an obstacle when scheduling an appointment? (to understand if during the scheduling process you had difficulties due to your language or ethnicity) | Yes; no; maybe |
| ¿Cree usted que su lengua materna es un obstáculo al momento de recibir la atención médica? (se busca comprender si en citas anteriores tuvieron dificultad para explicar su situación al especialista) | Si; no; tal vez | Do you think your native language is an obstacle when receiving medical care? (to understand if in previous appointments you had difficulty explaining your situation to the specialist) | Yes; no; maybe |

surveyed and asked to respond the two follow-up questions: "Was there any question that you did not understand?" and "Did any question seem confusing or ambiguous?" both with yes or no answers. When participants responded "Yes", they were asked to justify their answers. The responses were tabulated and analyzed using Pearson's Chi Square test and Z test for comparison of proportions. Table 3 describes the social, demographic and cultural characteristics of the participants in the validation.

A balance selection of participants was ensured to account for the cultural diversity across different regions in Ecuador. The results of both questions are presented in Tables 4 and 5. This type of validation was selected as the purpose of this questionnaire is to measure the waiting time in days and to identify whether people use the referral and counter-referral system stablished in the Ecuadorian Health Model, when knowing the origin of the medical appointment. Both measurement scales were classified as psychometric, that is, they can be measured objectively by asking the population directly [14].

Table 4 shows how participants from coastal region faced difficulties in understanding certain questions. When asked about the nature of these difficulties, they referred a problem concerning the identification of their group. Although, the terms *montuvio* or *mestizo*, are commonly used terms in Spanish, their association with the ethnicity could remain ambiguous, leading to this confusion. In contrast, respondents from the Andean highlands and

**Table 3. Characteristics of the population participating in the validation of the consistency of the questions in the questionnaire (n = 322).**

|  | N | % |
|---|---|---|
| **Total** | 322 | 100.0 |
| **Gender** | | |
| Female | 213 | 66.1 |
| Male | 109 | 33.9 |
| **Age group** | | |
| 18 to 25 years | 40 | 12.4 |
| 26 to 40 years | 124 | 38.5 |
| 41 to 55 years | 95 | 29.5 |
| 56 to 70 years | 53 | 16.5 |
| 71 years and older | 10 | 3.1 |
| **Income quintile** | | |
| Quintile 1 | 105 | 32.6 |
| Quintile 2 | 156 | 48.4 |
| Quintile 3 | 45 | 14 |
| Quintile 4 | 11 | 3.4 |
| Quintile 5 | 5 | 1.6 |
| **Education** | | |
| None | 14 | 4.3 |
| Primary | 99 | 30.7 |
| Secondary | 167 | 51.9 |
| Third level | 40 | 12.4 |
| Fourth level | 2 | 0.6 |
| **Area of residence** | | |
| Urban | 251 | 78.0 |
| Rural | 71 | 22.0 |
| **Ethnicity** | | |
| Mestizo | 299 | 92.9 |
| Indigenous | 12 | 3.7 |
| Afro-ecuadorian | 2 | 0.6 |
| Montuvio | 9 | 2.8 |
| **Language** | | |
| Spanish | 311 | 96.6 |
| Kichwua | 6 | 1.9 |
| Shuar | 5 | 1.6 |
| **Geographic region** | | |
| Andean highlands | 77 | 23.9 |
| Coastal | 110 | 34.2 |
| The Amazon | 135 | 41.9 |

Amazon regions, where a significant proportion of the population is indigenous and officially recognized in the country's political constitution, demonstrated a clearer understanding of their ethnic classification. To minimize respondent's errors related to this variable, the interviewers were instructed to provide a detailed explanation of the categorization.

Table 5 shows the responses to the questionnaire´s ambiguity. Since participants had previously received an explanation regarding ethnicity groups, the majority did not consider

**Table 4. Responses to the question "Were there any questions you did not understand?" from the people who participated in the validation of the consistency and understanding of the questions that made up the survey applied.**

| | Were there any questions you did not understand? (n 324) | | | | | | p-value |
|---|---|---|---|---|---|---|---|
| | No | | Yes | | Total | | |
| | N | % | N | % | N | % | |
| **Gender** | | | | | | | |
| Female | 188 a | 88.3 | 25 a | 11.7 | 213 | 100 | 0.131† |
| Male | 102 a | 93.6 | 7 a | 6.4 | 109 | 100 | |
| **Age group** | | | | | | | |
| 18 to 25 years | 36 a | 90 | 4 a | 10 | 40 | 100 | 0.344† |
| 26 to 40 years | 107 a | 86.3 | 17 a | 13.7 | 124 | 100 | |
| 41 to 55 years | 89 a | 93.7 | 6 a | 6.3 | 95 | 100 | |
| 56 to 70 years | 48 a | 90.6 | 5 a | 9.4 | 53 | 100 | |
| 71 years and older | 10 a | 100 | 0 a | 0 | 10 | 100 | |
| **Income quintile** | | | | | | | |
| Quintile 1 | 91 a | 86.7 | 14 a | 13.3 | 105 | 100 | 0.572‡ |
| Quintile 2 | 143 a | 91.7 | 13 a | 8.3 | 156 | 100 | |
| Quintile 3 | 42 a | 93.3 | 3 a | 6.7 | 45 | 100 | |
| Quintile 4 | 10 a | 90.9 | 1 a | 9.1 | 11 | 100 | |
| Quintile 5 | 4 a | 80 | 1 a | 20 | 5 | 100 | |
| **Education** | | | | | | | |
| None | 13 a | 92.9 | 1 a | 7.1 | 14 | 100 | 0.472‡ |
| Primary | 93 a | 93.9 | 6 a | 6.1 | 99 | 100 | |
| Secondary | 148 a | 88.6 | 19 a | 11.4 | 167 | 100 | |
| Third level | 34 a | 85 | 6 a | 15 | 40 | 100 | |
| Fourth level | 2 a | 100 | 0 a | 0 | 2 | 100 | |
| **Area of residence** | | | | | | | |
| Urban | 229 a | 91.2 | 22 a | 8.8 | 251 | 100 | 0.185† |
| Rural | 61 a | 85.9 | 10 a | 14.1 | 71 | 100 | |
| **Ethnicity** | | | | | | | |
| Mestizo | 267 a | 89.3 | 32 a | 10.7 | 299 | 100 | 0.434‡ |
| Indigenous | 12 a | 100 | 0 a | 0 | 12 | 100 | |
| Afro-ecuadorian | 2 a | 100 | 0 a | 0 | 2 | 100 | |
| Montuvio | 9 a | 100 | 0 a | 0 | 9 | 100 | |
| **Language** | | | | | | | |
| Spanish | 279 a | 89.7 | 32 a | 10.3 | 311 | 100 | 0.533‡ |
| Kichwua | 6 a | 100 | 0 a | 0 | 6 | 100 | |
| Shuar | 5 a | 100 | 0 a | 0 | 5 | 100 | |
| **Geographic region** | | | | | | | |
| Andean heghlands | 74 a | 96.1 | 3 b | 3.9 | 77 | 100 | 0.000† |
| Coastal | 85 a | 77.3 | 25 b | 22.7 | 110 | 100 | |
| The Amazon | 131 a | 97 | 4 b | 3 | 135 | 100 | |

Each letter in the subscript denotes a subset of the Were there any questions you didn't understand? categories whose column proportions do not differ significantly from each other at the .05 level.

†Refers to the p-value of Pearso's chi-square test

‡Refers to the p-value of Fisher's Exact Test

**Table 5. Responses to the question "Were any questions confusing or ambiguous?" from the people who participated in the validation of the consistency and understanding of the questions that made up the survey applied.**

| | Were any questions confusing or ambiguous? (n 324) | | | | | | p-value |
|---|---|---|---|---|---|---|---|
| | No | | Yes | | Total | | |
| | N | % | N | % | N | % | |
| **Gender** | | | | | | | |
| Female | 208 a | 97.7 | 5 a | 2.35 | 213 | 100.00 | 1.000‡ |
| Male | 106 a | 97.2 | 3 a | 2.75 | 109 | 100.00 | |
| **Age group** | | | | | | | |
| 18 to 25 years | 39 a | 97.5 | 1 a | 2.5 | 40 | 100.0 | 0.689‡ |
| 26 to 40 years | 119 a | 96.0 | 5 a | 4.0 | 124 | 100.0 | |
| 41 to 55 years | 94 a | 98.9 | 1 a | 1.1 | 95 | 100.0 | |
| 56 to 70 years | 52 a | 98.1 | 1 a | 1.9 | 53 | 100.0 | |
| 71 years and older | 10 a | 100.0 | 0 a | 0.0 | 10 | 100.0 | |
| **Income quintile** | | | | | | | |
| Quintile 1 | 102 a | 97.1 | 3 a | 2.9 | 105 | 100.0 | 0.408‡ |
| Quintile 2 | 152 a | 97.4 | 4 a | 2.6 | 156 | 100.0 | |
| Quintile 3 | 45 a | 100 | 0 a | 0 | 45 | 100.0 | |
| Quintile 4 | 10 a | 90.9 | 1 a | 9.1 | 11 | 100.0 | |
| Quintile 5 | 5 a | 100 | 0 a | 0 | 5 | 100.0 | |
| **Education** | | | | | | | |
| None | 14 a | 100.0 | 0 a | – | 14 | 100.0 | 0.789‡ |
| Primary | 97 a | 98.0 | 2 a | 2.0 | 99 | 100.0 | |
| Secondary | 161 a | 96.4 | 6 a | 3.6 | 167 | 100.0 | |
| Third level | 40 a | 100.0 | 0 a | – | 40 | 100.0 | |
| Fourth level | 2 a | 100.0 | 0 a | – | 2 | 100.0 | |
| **Area of residence** | | | | | | | |
| Urban | 245 a | 97.6 | 6 a | 2.4 | 251 | 100.0 | 0.838† |
| Rural | 69 a | 97.2 | 2 a | 2.8 | 71 | 100.0 | |
| **Ethnicity** | | | | | | | |
| Mestizo | 291 a | 97.3 | 8 a | 2.7 | 299 | 100.0 | 1.000‡ |
| Indigenous | 12 a | 100.0 | 0 a | 0.0 | 12 | 100.0 | |
| Afro-ecuadorian | 2 a | 100.0 | 0 a | 0.0 | 2 | 100.0 | |
| Montuvio | 9 a | 100.0 | 0 a | 0.0 | 9 | 100.0 | |
| **Language** | | | | | | | |
| Spanish | 303 a | 97.4 | 8 a | 2.6 | 311 | 100.0 | 1.000‡ |
| Kichwua | 6 a | 100.0 | 0 a | 0.0 | 6 | 100.0 | |
| Shuar | 5 a | 100.0 | 0 a | 0.0 | 5 | 100.0 | |
| **Geographic region** | | | | | | | |
| Andean highlands | 74 a | 96.1 | 3 a | 3.9 | 77 | 100.0 | 0.664‡ |
| Coastal | 108 a | 98.2 | 2 a | 1.8 | 110 | 100.0 | |
| The Amazon | 132 a | 97.8 | 3 a | 2.2 | 135 | 100.0 | |

Each letter in the subscript denotes a subset of the Were there any questions you didn't understand? categories whose column proportions do not differ significantly from each other at the .05 level.

† Refers to the p-value of Pearson's chi-square test

‡ Refers to the p-value of Fisher's Exact Test

the questionnaire confusing. This provided the necessary confidence to conclude that the explanation given by interviewers would be sufficient to minimize errors, especially among the coastal region participants.

## Data management

All data will be securely stored in password-protected files, and access will be limited to the research team. Personal identifiers will not be collected ensuring participants' confidentially.

## Data analysis

The data will be handled as follows: interviewers will collect the information using a a Google Forms survey software, which allows information to be exported as an Excel (Microsoft Office) matrix, then to be tabulated and coded. The coded matrix will be processed with IBM SPSS version 25. In this database, frequency distribution tables will be generated and the access methods will be analyzed in relation to waiting time.

## Statistical analysis

Statistical analysis will include the calculation of means and standard deviations for continuous variables and the calculation of frequencies and percentages for categorical variables. For the statistical analysis, it is intended to make a comparison of the average time of specialized medical care, both for an appointment with a specialist and for the procedure, to then relate it to the social, demographic, economic variables and especially with the way in which the user accessed the appointment. To do so, the normality of the Waiting Time variable will be tested with the Kolmogorov-Smirnov test. If the result is normal, parametric tests such as ANOVA (post hoc Tukey), T test and Pearson correlations will be used depending on the independent variable to be analyzed. If the result is not normal, Kruskal-Wallis, Mann-Whitney and Spearman will be used. Additionally, a linear regression model will be created taking Waiting Time as the dependent variable and the Access Methods as the independent variable.

The variable Access Method will be the independent variable for the analyses mentioned above. In addition, the response options will be grouped to form a new dichotomous variable "Standardized Access" with categories of Yes and No. To do this, the response options to the variable Form of Access: Through a friend or family member and through a person who is not a friend or family member, will be considered as Non-Standardized and the options: through a referral from a doctor at a health center, through a specialist at this or another hospital and through an appointment at the hospital or by phone call, will be considered as Yes Standardized.

This variable will become dependent in order to associate it with the educational level, economic level and area of residence and determine whether belonging to any category of these variables increases the chances of entering in a standardized way or not. For the latter, it is intended to use a logistic regression and obtain OR as a measure of association. For all tests, a significance level of 0.05 will be used as a reference. The statistical program SPSS version 25 will be used.

## Ethics statement

This research project was reviewed and approved by the Ethics Committee for Research on Human Beings of ITSUP with the number 1718079732. The ethics committee is registered in the Office for Human Research Protections with the number RB00014260.

All users who participate in this research will have the possibility of refusing to participate. Those who agree to do so will be informed through an informed consent, about the objective of the study. This consent was approved by the Ethics Committee that approved the protocol

and will be in the main part of the response registration form. The consent form will be digital and to approve participation in the study, participants must press the accept button to start the questions. Only people over 18 years of age will participate, so informed consent will not be necessary for minors. A copy of the consent form will be sent to users at the email addresses they provide for this purpose. Even if they have agreed to participate, people can stop doing so at any time after the survey has started. In that case, their registration will be deleted in the presence of the participant. Participants will not receive any compensation for their participation. In addition, they will be informed of an email address where they can request more information about the research if they require it.

The interviewers will be properly identified and will respect the decision of the participants. They will not ask to see the identification or ask about the medical problems that led users to request an appointment at the hospital, only about the specialty they attend. Since the diseases treated by each medical specialty are very varied, it is not possible to determine the specific medical condition of a person just by knowing what type of specialist they attend.

This protocol was written and conditioned at Protocols.io and can be reviewed at: https://dx.doi.org/10.17504/protocols.io.261ge5z7wg47/v1

## Justification

Investigating waiting times and methods of access to specialized health services in public hospitals in Ecuador is crucial for several reasons. Waiting times for specialized medical consultations in Ecuador's public hospitals can be significantly long, averaging 49 days, with some cases extending up to 180 days. These prolonged waiting periods can have serious implications for patient health outcomes and the overall efficiency of the healthcare system. Briones et al (2024) highlighted the existence of non-standardized methods to obtain specialized medical appointments in public hospitals in Manabí, Ecuador. Patients who accessed appointments through informal means (e.g., with help from family or friends working at the hospital) experienced shorter waiting times by up to 19 days compared to those who used formal procedures [8]. While informal access has been mentioned anecdotally, its potential role in exacerbating inequities within the health system has yet to be systematically studied in Ecuador.

Non-standardized access to the healthcare system may perpetuate and even increase social inequalities, especially in Latin America, where structural inequities deeply influence living conditions and health outcomes [15]. Socioeconomic disparities are key drivers of health access inequalities, and frequent barriers include the ability to pay, geographic distance, availability of services, cultural or ethnic differences, communication challenges, and architectural obstacles [15]. Unregulated access could potentially worsen these issues by allowing individuals with more resources or connections to access healthcare more quickly, leaving marginalized populations at a disadvantage. Despite various Latin American countries implementing health reforms since the 1990s aimed at strengthening healthcare systems and reducing inequalities in access and outcomes [16], significant disparities persist. This suggests that unregulated access, without adequate oversight, may not address health inequalities and could worsen them in the absence of targeted interventions.

Addressing these health inequalities effectively requires the implementation of both structuring and sectorial policies in Latin America [15]. These policies must regulate access to healthcare while also addressing the social determinants of health, such as income, education, transportation, and living conditions. Furthermore, a human rights-based approach, which includes principles of accountability, meaningful participation, transparency, and non-discrimination, should be incorporated into health policies to promote equitable universal health coverage in the region [17].

Ecuador has a complex healthcare system, serving a heterogeneous population of almost 18 million people through both public and private hospitals [18]. This protocol aims to provide valuable information on waiting times and access methods to specialized services, contributing to the planning of more equitable health policies. Investigating waiting times and methods of access could help identify areas for improvement in these essential healthcare services. Studying these factors is critical for addressing healthcare inequities, improving service delivery, and ultimately enhancing patient outcomes in a healthcare system that faces significant resource limitations. To do so, we are proposing a methodology that is flexible and can be adapted for use in other countries with similar health systems, potentially facilitating international comparisons in future studies.

After an exhaustive search of the scientific literature, we found no studies on how Ecuadorians access specialized health services. Although informal access to specialized services has been mentioned anecdotally, it has not been formally studied in Ecuador. This study seeks to determine whether this practice occurs and how it influences waiting times, providing crucial insights for health policy.

The *Instituto Nacional de Estadística y Censos* (INEC), through its national health and nutrition survey, collects data on waiting times, but only for the period after a patient arrives at the hospital for treatment [13]. It does not capture the waiting time between obtaining an appointment and receiving care. This study will expand on this variable, providing new information for public health planning. Additionally, identifying which medical specialties have the longest waiting times could reveal gaps in the availability of specialists, informing the need for targeted interventions.

The choice to use surveys to collect data on waiting times mirrors the methodology employed by INEC, Ecuador's official statistical authority. The questions and response options in this study are similar to those used by INEC but have been modified to include information that INEC does not cover, thereby addressing the specific research question. This approach ensures reliability in the data collected. The use of trained surveyors will also improve the accuracy of responses by ensuring participants fully understand the survey's objectives. Conducting the survey digitally and providing a link to users was considered but posed a significant risk of non-response or inaccurate data, especially among populations with limited internet access.

As described in the methodology, the survey is not designed to diagnose a condition or form a construct. Instead, it seeks to gather objective data on waiting times and access methods. Therefore, it was not necessary to validate the questionnaire as a tool for measuring relationships between multiple variables. Instead, we validated the clarity and comprehensibility of the questions and response options. As detailed in the methodology, the survey questions were well-understood across the regions surveyed. In areas like the coastal region of Ecuador, where participants do not commonly group themselves by ethnicity, the role of trained surveyors becomes especially important. These surveyors will explain any potentially unfamiliar terms, such as "ethnicity," to ensure consistent understanding across regions.

The questionnaire was designed using questions previously validated in national studies, but it has been adapted to include new variables specific to this research. A comprehension test was conducted in different regions of Ecuador to ensure that all respondents fully understood the questions. Waiting for patients outside hospitals will allow us to reach a broad range of participants, including those without internet access, the elderly who may have difficulties using technology, and individuals with disabilities. This approach will ensure equitable participation from a representative sample of the Ecuadorian healthcare system's users. The survey will be conducted in multiple hospitals in the most densely populated provinces, such as Guayas, Pichincha, and Manabí. In other provinces, we plan to visit the single general or regional hospital available in each location.

It is planned to publish the results of the research in the same journal that agrees to publish the protocol.

## Limitations

This study presents some limitations. First, it did not include basic and highly specialized hospitals. However, this omission does not affect the validity of our findings, since these facilities were not considered in the study design. Secondly, although the sample was stratified based on the number of patients attending hospitals within a given reference period, stratification by socioeconomic status was no feasible. Also, since the Ministry of Public Health hospitals provide service free of charge, the present study will only captures access methods and the waiting time within a specific population, which may not be representative of the entire Ecuadorian population.

Additionally, the variable waiting time between scheduling an appointment and the actual day of care relies on participants to remember the exact day of their appointment. This reliance introduces a potential recall bias, particularly among those who have longer waiting periods. Finally, regarding access methods, there is the possibility that participants may not disclose the true means by which they obtained their appointments, especially if they benefited from non-standardized access methods. To mitigate the bias, interviewers will emphasize the confidentiality of their answers throughout data collection.

## Dissemination plans

The protocol and the results of the research will be disseminated through publications in scientific journals. In the event of modifications during the protocol review process, these will be adapted to the questionnaire and the data collection procedure.

## Supporting information

**S1 File. The database for the SPSS statistical program is used as support for the data analysis carried out for the evaluation of the pilot study.**
(SAV)

## Acknowledgments

We would like to thank the Research Center of the University Espíritu Santo for their trust and support in the creation of this research protocol.

To receive the approval from the ethics committee, it was required to present a letter of interest from the institution where the research was to be conducted. The Ministry of Public Health did not provide the aforementioned letter; therefore, all data collection will be carried out outside the hospitals. As it is a public space, approval from this institution is not necessary.

## Author contributions

**Conceptualization:** Marcelo Armijos Briones, Patricia Estefanía Ayala Aguirre.

**Data curation:** Marcelo Armijos Briones, Sammy Figueroa Intriago.

**Formal analysis:** Antonio Lanata-Flores.

**Investigation:** Marcelo Armijos Briones, Sammy Figueroa Intriago, Oscar Marcillo Toala, Patricia Estefanía Ayala Aguirre.

**Methodology:** Pablo Benitez Sellán, Patricia Estefanía Ayala Aguirre.

**Project administration:** Oscar Marcillo Toala.

**Resources:** Sammy Figueroa Intriago, Pablo Benitez Sellán.

**Supervision:** Marcelo Armijos Briones, Oscar Marcillo Toala.

**Validation:** Marcelo Armijos Briones.

**Writing – original draft:** Marcelo Armijos Briones.

**Writing – review & editing:** Antonio Lanata-Flores, Pablo Benitez Sellán, Patricia Estefanía Ayala Aguirre.

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
