## [Decision Letter · Decision Letter 0]

26 Dec 2024

PONE-D-24-48992Protocol: Waiting time and ways of accessing specialized health services in public hospitals in Ecuador.PLOS ONE

Dear Dr. Armijos Briones,

Thank you for submitting your manuscript to PLOS ONE. After careful consideration, we feel that it has merit but does not fully meet PLOS ONE’s publication criteria as it currently stands. Therefore, we invite you to submit a revised version of the manuscript that addresses the points raised during the review process.

Please address the methodological issues raised by the reviewers, both in terms of the potential bias on the variables, and in terms of the appropriateness of the proposed analytical model.

We look forward to receiving your revised manuscript.

Kind regards,

Juan Pablo Gutierrez

Academic Editor

PLOS ONE

Journal Requirements:

Please confirm at this time whether or not your submission contains all raw data required to replicate the results of your study. Authors must share the “minimal data set” for their submission. PLOS defines the minimal data set to consist of the data required to replicate all study findings reported in the article, as well as related metadata and methods (https://journals.plos.org/plosone/s/data-availability#loc-minimal-data-set-definition ).

If your submission does not contain these data, please either upload them as Supporting Information files or deposit them to a stable, public repository and provide us with the relevant URLs, DOIs, or accession numbers. For a list of recommended repositories, please see https://journals.plos.org/plosone/s/recommended-repositories .

Reviewers' comments:

Reviewer's Responses to Questions

**Comments to the Author**

1. Does the manuscript provide a valid rationale for the proposed study, with clearly identified and justified research questions?

Reviewer #1: Yes

Reviewer #2: Yes

2. Is the protocol technically sound and planned in a manner that will lead to a meaningful outcome and allow testing the stated hypotheses?

Reviewer #1: Partly

Reviewer #2: Yes

3. Is the methodology feasible and described in sufficient detail to allow the work to be replicable?

Reviewer #1: Yes

Reviewer #2: Yes

4. Have the authors described where all data underlying the findings will be made available when the study is complete?

Reviewer #1: Yes

Reviewer #2: Yes

5. Is the manuscript presented in an intelligible fashion and written in standard English?

Reviewer #1: Yes

Reviewer #2: Yes

6. Review Comments to the Author

You may also provide optional suggestions and comments to authors that they might find helpful in planning their study.

Reviewer #1: In general, the protocol meets the required publication criteria, has an identified and justified research question, has current approval from an ethics committee, declares funding, includes preliminary results of a pilot test, has not yet been data collected for the study, and the place where these can be consulted publicly once they are available has been contemplated, however, some important points should be worked on before being published.

1. The variable “waiting time” is central to the study; previous publications have obtained this information from hospital records (1, 2), but this protocol proposes asking patients directly without verifying the data, which has the risk that the data is biased by the patients' memory: Is this bias will be controlled in some way? This is relevant since it has been reported that patients tend to overestimate the waiting time (3) which could lead to biased conclusions. The recommendation is to at least mention this bias as a limitation, taking into account its implication in the conclusions.

2. The other variable of interest “Form of access to the hospital” is subject to information bias since patients could lie about it when their entry has not been through a “standardized” route; this is controlled by emphasizing the confidentiality of the data to the people interviewed, however, it is a latent bias of which there is no certainty that it can be eliminated, so it must be mentioned in the limitations with the actions taken to reduce it.

3. The proposed statistical model does not seem to be correct. In the analysis section (page 17) it is mentioned that the variable “Form of Access” will be taken as dependent, however, how the study is planned, the “Form of Access” is the independent variable that is expected to influence in the “waiting time” which is the dependent variable. Therefore, a more appropriate model than the logistic one would be multiple linear regression, in which the other sociodemographic variables of interest could be included, as long as the assumptions of the model are met.

4. Regarding the questionnaire, the objective of asking questions 31 and 32 is not clear. They are sensitive questions that could make the person interviewed uncomfortable and do not seem to be directly related to the project, so it is recommended that the reason why they were included be justified, or otherwise discarded from the questionnaire.

5. It is not clear if section 3 of the questionnaire was also piloted. As it is the section that contains the variables of greatest interest in the study, it is recommended to pilot it, and if it has already been done, it is recommended to report the findings found.

6. General comments: On page 2, paragraph 2, line 8, the phrase “According to the results” is repeated; A protocol does not have a discussion section since results have not yet been obtained, so it is recommended to change the subtitle on page 19 to “Justification.”

References

1. Zhang, Hui PhDa; Ma, Weimin PhDa,*; Zhou, Shufen MDb; Zhu, Jingjing MDc; Wang, Li MDd; Gong, Kaixin MDa. Effect of waiting time on patient satisfaction in outpatient: An empirical investigation. Medicine 102(40):p e35184, October 06, 2023. | DOI: 10.1097/MD.0000000000035184

2. Ogawa T, Tachibana T, Yamamoto N, Udagawa K, Kobayashi H, Fushimi K, Yoshii T, Okawa A, Jinno T. Patient body mass index modifies the association between waiting time for hip fracture surgery and in-hospital mortality: A multicenter retrospective cohort study. J Orthop Sci. 2022 Nov;27(6):1291-1297. doi: 10.1016/j.jos.2021.07.015. Epub 2021 Aug 13. PMID: 34393026.

3. Thompson DA, Yarnold PR, Adams SL, Spacone, AB: How accurate are waiting time perceptions of patients in the emergency department? Ann Emerg Med 1996 28:652-656

Reviewer #2: Suggestion:

In the background section, it would be helpful to the readers if it briefly describes the health system in Ecuador.

Minor observations:

In the abstract, the last sentence repeats, "According to the results."

Table 2 the answer to the variable "sex," It would be better to use "hombre y mujer."

7. PLOS authors have the option to publish the peer review history of their article (what does this mean? ). If published, this will include your full peer review and any attached files.

**Do you want your identity to be public for this peer review?** For information about this choice, including consent withdrawal, please see our Privacy Policy .

Reviewer #1: No

Reviewer #2: **Yes: ** Samuel Gutierrez-Barreto

---

## [Author Response · Author response to Decision Letter 1]

11 Feb 2025

Journal Requirements:

Please ensure that your manuscript meets PLOS ONE's style requirements, including those for file naming. The PLOS ONE style templates can be found at https://journals.plos.org/plosone/s/file?id=wjVg/PLOSOne_formatting_sample_main_body.pdf and https://journals.plos.org/plosone/s/file?id=ba62/PLOSOne_formatting_sample_title_authors_affiliations.pdf

A: The guidelines were carefully reviewed and followed throughout the manuscript.

A: In the methodology section, in the section on ethical considerations, it was stated that the informed consent that will be used is digital, it will be shown and read to the participant and, only when he approves it by clicking on the accept button, will the survey proceed. People under 18 years of age will not participate, therefore, informed consent will not be necessary. A copy of the informed consent will be uploaded as metadata.

A: As a supplementary file, we will upload the original database that was used for the validation of the questionnaire to be used in .sav format for the SPSS program.

6. Review Comments to the Author

Reviewer #1: In general, the protocol meets the required publication criteria, has an identified and justified research question, has current approval from an ethics committee, declares funding, includes preliminary results of a pilot test, has not yet been data collected for the study, and the place where these can be consulted publicly once they are available has been contemplated, however, some important points should be worked on before being published.

The variable “waiting time” is central to the study; previous publications have obtained this information from hospital records (1, 2), but this protocol proposes asking patients directly without verifying the data, which has the risk that the data is biased by the patients' memory: Is this bias will be controlled in some way? This is relevant since it has been reported that patients tend to overestimate the waiting time (3) which could lead to biased conclusions. The recommendation is to at least mention this bias as a limitation, taking into account its implication in the conclusions.

A: We consider this observation to be very valid, we appreciate and accept it. We have placed the memory bias in the limitations section, and we will be cautious with our conclusions when we have the results of our research.

The other variable of interest “Form of access to the hospital” is subject to information bias since patients could lie about it when their entry has not been through a “standardized” route; this is controlled by emphasizing the confidentiality of the data to the people interviewed, however, it is a latent bias of which there is no certainty that it can be eliminated, so it must be mentioned in the limitations with the actions taken to reduce it.

A: We consider this observation to be very valid, we appreciate and accept it. We have placed the bias of possible dishonest information in the limitations section and we will emphasize confidentiality during data collection.

3. The proposed statistical model does not seem to be correct. In the analysis section (page 17) it is mentioned that the variable “Form of Access” will be taken as dependent, however, how the study is planned, the “Form of Access”

is the independent variable that is expected to influence in the “waiting time” which is the dependent variable. Therefore, a more appropriate model than the logistic one would be multiple linear regression, in which the other sociodemographic variables of interest could be included, as long as the assumptions of the model are met.

A: The observation is correct, our dependent variable is the waiting time and our main analysis will be to determine how the Access Method (independent variable) affects the waiting time. However, we consider that it is important to analyze what part of the population can use non-standard scheduling. For this, the Access Method variable will be used to create a new variable "Standard scheduling" with the categories Yes and No, this will allow us to use this variable as a dependent variable to perform a logistic regression and determine who are the people who have more opportunity to enter in a non-standard way. We have explained this better in the statistical analysis section.

In addition, we have welcomed the suggestion of returning a linear regression model with our main variables.

4. Regarding the questionnaire, the objective of asking questions 31 and 32 is not clear. They are sensitive questions that could make the person interviewed uncomfortable and do not seem to be directly related to the project, so it is recommended that the reason why they were included be justified, or otherwise discarded from the questionnaire.

A: We consider this observation to be very valid, we appreciate it and accept it. We have eliminated these two questions from our questionnaire.

5. It is not clear if section 3 of the questionnaire was also piloted. As it is the section that contains the variables of greatest interest in the study, it is recommended to pilot it, and if it has already been done, it is recommended to report the findings found.

A: The pilot study served to demonstrate the understanding of the questions by the users. In this sense, all the sections with their respective questions were part of the pilot. We considered not including the data obtained from our main variables because they could be considered as preliminary data without having yet approved the protocol article.

6. General comments: On page 2, paragraph 2, line 8, the phrase “According to the results” is repeated; A protocol does not have a discussion section since results have not yet been obtained, so it is recommended to change the subtitle on page 19 to “Justification.”

A: The repeated phrase was eliminated. The Discussion section was changed to Justification.

Reviewer #2: Suggestion:

In the background section, it would be helpful to the readers if it briefly describes the health system in Ecuador.

A: In the first paragraph of our introduction, a brief description of how the Ecuadorian health system works was placed.

Minor observations:

In the abstract, the last sentence repeats, "According to the results."

Table 2 the answer to the variable "sex," It would be better to use "hombre y mujer."

A: The repeated phrase was eliminated. The name Masculine and Feminine was changed to Man or Woman, both in the manuscript and in the questionnaire to be used.

---

## [Decision Letter · Decision Letter 1]

9 Mar 2025

Protocol: Waiting time and ways of accessing specialized health services in public hospitals in Ecuador.

PONE-D-24-48992R1

Dear Dr. Armijos Briones,

We’re pleased to inform you that your manuscript has been judged scientifically suitable for publication and will be formally accepted for publication once it meets all outstanding technical requirements.

Kind regards,

Juan Pablo Gutierrez

Academic Editor

PLOS ONE

Additional Editor Comments (optional):

Reviewers' comments:

Reviewer's Responses to Questions

**Comments to the Author**

1. Does the manuscript provide a valid rationale for the proposed study, with clearly identified and justified research questions?

Reviewer #1: Yes

Reviewer #2: Yes

2. Is the protocol technically sound and planned in a manner that will lead to a meaningful outcome and allow testing the stated hypotheses?

Reviewer #1: No

Reviewer #2: Yes

3. Is the methodology feasible and described in sufficient detail to allow the work to be replicable?

Reviewer #1: Yes

Reviewer #2: Yes

4. Have the authors described where all data underlying the findings will be made available when the study is complete?

Reviewer #1: Yes

Reviewer #2: Yes

5. Is the manuscript presented in an intelligible fashion and written in standard English?

Reviewer #1: Yes

Reviewer #2: Yes

6. Review Comments to the Author

You may also provide optional suggestions and comments to authors that they might find helpful in planning their study.

Reviewer #1: In general, the protocol meets the required publication criteria, has an identified and justified research question, has current approval from an ethics committee, declares funding, includes preliminary results of a pilot test, has not yet been data collected for the study, and the place where these can be consulted publicly once they are available has been contemplated. Also, the previous comments have been satisfactorily answered by the authors.

Reviewer #2: All the suggestions have been addressed. I have no more suggestions on this protocol.

7. PLOS authors have the option to publish the peer review history of their article (what does this mean? ). If published, this will include your full peer review and any attached files.

**Do you want your identity to be public for this peer review?** For information about this choice, including consent withdrawal, please see our Privacy Policy .

Reviewer #1: No

Reviewer #2: No

---

## [Editor Report · Acceptance letter]

PONE-D-24-48992R1

PLOS ONE

Dear Dr. Armijos Briones,

I'm pleased to inform you that your manuscript has been deemed suitable for publication in PLOS ONE. Congratulations! Your manuscript is now being handed over to our production team.

Kind regards,

on behalf of

Dr. Juan Pablo Gutierrez

Academic Editor

PLOS ONE